# Comparison of Tribological Characteristics of AA2024 Coated by Plasma Electrolytic Oxidation (PEO) Sealed by Different sol–gel Layers

Hafiza Ayesha Khalid [1,2], Sajjad Akbarzadeh [1,2], Yoann Paint [3], Véronique Vitry [2] and Marie-Georges Olivier [1,3,*]

1 Materials Science Department, Faculty of Engineering, University of Mons, 20, Place du Parc, 7000 Mons, Belgium; hafizaayesha.khalid@umons.ac.be (H.A.K.)
2 Metallurgy Department, Faculty of Engineering, University of Mons, 20, Place du Parc, 7000 Mons, Belgium; veronique.vitry@umons.ac.be
3 Materia Nova Research Centre, 1, Avenue N. Copernic, Parc Initialis, 7000 Mons, Belgium
* Correspondence: marjorie.olivier@umons.ac.be

**Abstract:** The application of sol–gel on plasma electrolytic oxidation (PEO) coatings can increase wear resistance by sealing the surface defects such as pores and cracks in the outer layer of the PEO layer and strengthen the coating. Four different sol–gel formulations based on precursors—(3-glycidyloxypropyl)trimethoxysilane (GPTMS), methyltriethoxysilane (MTES), methacryloxypropyltrimethoxysilane (MAPTMS), (3-aminopropyl)triethoxysilane (APTES), and zirconium(IV) propoxide (ZTP) along with tetraethoxysilane (TEOS)—were used to seal PEO pores, and the samples were tested tribologically. A sliding reciprocating tribometer was used to carry out a wear test with an alumina ball as the counter body in two different conditions: (a) 2.5 N load for 20 min, and (b) 3 N load for 40 min. The coefficient of friction and wear rate as volume loss per unit sliding length were obtained for all sol–gel-sealed specimens and unsealed PEO-coated and bare AA2024 substrate. 3D mechanical profilometer surface scans were used to compare the depth of wear traces. The elemental color mapping using SEM and EDS revealed that silicon remains present in the wear tracks of PEO coatings sealed with sol–gel layers containing GPTMS (PSG) and ZTP (PSG-ZT). GPTMS (PSG) was able to fill the pores of the PEO layer efficiently due to its cross-linked network. Moreover, sol–gel containing ZTP (PSG-ZT) was deposited as a thick layer on top of the PEO layer which provided good lubrication and resistance to wear. However, other sol–gel formulations (PSG-MT and PSG-AP) were worn out during tests at a higher load (3 N). The most stable friction coefficient (COF) and specific wear rates were observed with sol–gels with GPTMS and ZTP.

**Keywords:** plasma electrolytic oxidation; sol–gel sealing; AA2024; wear; SEM; profilometer





## 1. Introduction

Surface degradation by wear has recently been catching attention due to the frequent need of replacing parts in the engineering industry. This phenomenon is disastrous, leading to performance and energy losses that occur by a worn surface of the counter component when two bodies are in relative motion. The choice of material should fulfill certain criteria which are defined based on the mechanical properties, durability, mass properties, cost of production, and ease of fabrication [1]. With rising environmental concerns, lighter-weight materials with reliable strength are in focus, to be used for components and assemblies under a motion to save fuel costs. To fulfill this objective, aluminum alloys, owing to their low density and high specific resistance, have played a vital role in replacing conventional construction materials (steel) [2,3]. Nevertheless, bulk material with a hard surface that can withstand mechanical and tribological loads as well as exhibit high resistance to corrosion cannot be accomplished from aluminum alloys in their natural form. Aluminum alloys form

a soft, thin oxide layer that is easily removed when subjected to corrosive and tribological loads. Plasma electrolytic oxidation (PEO) is an electrochemical process that produces a crystalline oxide layer at a metallic substrate by generating a plasma discharge at the metal–electrolyte interface. Several-microns-thick PEO coatings reach a high hardness and present excellent adhesion to the substrate. Mora-Sanchez et al. found a significant increase in the hardness of PEO-treated cast A361 alloy—up to ~15 GPa—compared to hard anodized coatings on the same alloy which yielded a hardness of only up to ~4 GPa [4]. The PEO process is the method of choice that utilizes environmentally friendly materials such as diluted alkaline electrolytes [5–7]. Such coatings provide good resistance to fretting, abrasion, and erosion by improving the wear resistance of aluminum alloys [8–10].

There are, however, some limitations with PEO coatings, as the structure contains micropores throughout the PEO layers which results from the micro-arc generated by plasma on the metal–electrolyte interface. Moreover, the coating growth occurs with the simultaneous melting and solidification, leading to thermal stresses which generate micro-cracks radially propagated from the pores [11]. The source of power mode employed in the PEO process has an impact on the hardness, growth rate, phase composition, structure, morphology, and degree of porosity of the coatings. The PEO treatment operating in DC mode results in coatings with a lower oxide development rate and higher porosity due to its restricted control and difficulty in changing discharge characteristics. Nevertheless, the pulsed DC mode provides the opportunity to regulate the discharge duration and can potentially use less energy [12]. By using the AC mode, electrode polarization is avoided, and arc interruption may be used to conveniently manage the process. In comparison to coatings created using DC, AC, and unipolar pulsed modes, the bipolar pulsed mode greatly enhances coating characteristics, drawing the attention of numerous researchers. Since it lessens the frequency of strong plasma discharges and high-temperature spikes during the PEO process, dense coatings with superior corrosion resistance and larger coating thickness may be created in the bipolar pulsed current mode [13,14]. Under AC or bipolar-pulsed regimes, frequently using frequencies up to several thousand Hz and variable duty cycles, a three-layered PEO coating consisting of an outer porous layer rich in electrolyte-derived compounds, an intermediate relatively compact micro-cracked layer rich in $\alpha$-alumina, and a submicron dense barrier layer can be obtained [15]. According to theory, increasing the quantity of $\alpha$-$Al_2O_3$ improves the wear performance of PEO coatings [16]. Khan et al. [17] discovered that a reduced duty cycle resulted in a comparable decline in the ratio of $\alpha$-$Al_2O_3$ to $\gamma$-$Al_2O_3$ in PEO coatings on 6082 aluminum alloy generated by the pulsed unipolar current. The concentration of $\alpha$-$Al_2O_3$ in the coatings was observed to rise with the application of larger current densities and the lengthening of the deposition period, which produced thicker coatings [18,19]. V. Dehnavi et al. [20] found that increasing the pulse on time by employing a lower frequency and higher duty cycle generates micro discharges with a lower spatial density but higher intensity which results in higher concentrations of Si-rich species on the surface of the PEO coatings. V. Hutsaylyuk et al. [21] reduced the unfavorable impact of hydrogen on the plasma-electrolytic oxidation of aluminum alloys and accordingly increased the efficiency of the synthesis of PEO layers with high abrasive-wear resistance. In general, high frequencies and negative pulses help to limit the duration of individual micro discharges, preventing their reappearance at the same location and transition into destructive arc discharges [22].

Among the several post-treatments to seal PEO pores, sol–gel has been described as an efficient sealing treatment for pores to avoid the penetration of aggressive media inside the pores [23,24]. It acts as a cementing agent in the PEO layer and can provide good resistance to corrosion and wear [25]. Since the benefit of using this process allows the composition of the sol–gel to be adjusted, the surfaces can therefore be tailored to the application [26,27]. Sopchenski-Santos et al. [28] developed PEO coatings on AA2024 and sealed them with sol–gel based on silicon precursors. They observed improvement in the wear resistance of the sol–gel-sealed PEO specimens over unsealed PEO-coated and bare substrate. In the

previous studies carried out by our research group, a significant improvement in resistance to corrosion has been noticed under familiar cases [29].

Sol–gel properties are influenced by several variables, including solvent type, aging, solution pH, and the type of sol–gel precursor, particularly [30,31]. In our previous investigation, four types of sol–gel solutions using Si and Si/Zr-based precursors with different organic functional groups (epoxy, amine, and methyl) have been prepared. The hydrophobicity, compactness, and impregnation characteristics of the sol–gel layers could dramatically affect the corrosion resistance properties. It was explained that the addition of MTES reduced the cross-linking density and compactness of a sol–gel layer consisting of TEOS and GPTMS precursors due to the presence of one un-hydrolysable methyl group. GPTMS could provide pore-filling properties, leading to the identical sealing ability of two types of sol–gel coatings. It was described that the sol–gel composed of APTES and TEOS precursors had the worst corrosion protection performance due to the insufficient sealing ability and relatively hydrophilic properties. The best anti-corrosion properties were observed in a PEO/sol–gel coating system in which TEOS, ZTP, and MAPTMS were employed as sol–gel precursors. Relatively high hydrophobicity, the capability to form a uniform layer over PEO as well as the ability to penetrate intrinsic pores, and the evolution of the cage-like siloxane network along with the ladder-like structure were the crucial factors leading to the best sealing abilities [32]. This research aims to investigate the tribological behavior of PEO-coated AA2024 alloy by sealing the PEO pores with various sol–gel formulations. To the best of the author's knowledge, no investigation has been made on the tribological behavior of sol–gel-sealed PEO coatings on AA2024 alloy with different Si- and Si/Zr-based sol–gel formulations. In the continuation of our previous work [32], a tribological evaluation is carried out with the same sol–gel formulations used to seal the PEO layer, and PEO conditions were also kept identical.

## 2. Experimental Procedure

AA2024 alloy (Ti $\leq$ 0.15%, Ni 0.15%, Zn 0.25%, Si 0.5%, Fe 0.5%, Mn 0.3%–0.9%, Mg 1.2%–1.8%, Cu 3.8%–4.9%, and Al balance) specimen with dimensions 35 mm $\times$ 25 mm $\times$ 1.5 mm were used in this study to produce PEO layer with parameters previously defined by our research group [17,21]. A PowerPulse (Micronics Systems, Vilette d'Anthon, France) was used to coat specimens for 30 min using the following conditions: bipolar regime, 5A anodic current, 30% duty cycle, 100 Hz frequency, in an alkaline electrolytic bath consisting of 1.65 g/L $Na_2SiO_3$ and 1 g/L KOH (Alfa Aesar Co., Tewksbury, MA, USA). Four (4) different sol–gel formulations previously prepared by our research group were used to seal the PEO pores, the details of which can be found in an article by Akbarzadeh et al. [32]. The details of sol–gel formulations are given in Table S1. To briefly bring up the various sol–gel formulations, SG included TEOS (20% $v/v$) and GPTMS (10% $v/v$) in an electrolyte composed of distilled water (60% $v/v$) and ethanol (10% $v/v$). After bringing the pH down to 3, the solution was stirred for 24 h. The same condition was utilized to prepare PSG-MT but with one change, in which the concentration of TEOS decreased to 10% $v/v$ and MTES with the concentration of 10% $v/v$ was added instead to attain 30% $v/v$ overall precursor concentration. These two solutions were applied to the PEO-coated samples, followed by curing at 150 °C for one hour to attain PSG and PSG-MT samples. To achieve the overall percentage of 30% $v/v$ silane precursors in a solution including ethanol (56% $v/v$) and distilled water (14% $v/v$) in another sol–gel solution (SG-AP), a comparable volumetric intake of APTES (15% $v/v$) and TEOS (15% $v/v$) was used. Following this, acetic acid was gradually added to the solution to maintain the pH at 4.5. The mixture was stirred for a day at room temperature. The PEO sample coated with SG-AP was placed at ambient temperature for one day followed by placing at 150 °C for one hour to obtain PSG-AP. For the SG-ZT, two types of solution were mixed. The initial solution (Sol 1) was made up of TEOS (0.18 mol) and MAPTMS (1 mol) precursors, and the hydrolysis and condensation processes were started by adding distilled water (2.075 mol) and hydrochloric acid dropwise (0.001 mol). It was then added dropwise to Sol 2 (ZTP (0.12 mol), MAA (0.12 mol), and

isopropyl alcohol (0.4 mol)) after 150 min of stirring, then the mixture was stirred for a full day. The coated panels were dried with SG-ZT solution for one hour at 100 °C to attain the PSG-ZT sample. The sol–gel formulations were coated onto the PEO specimens through KSV Nima dip-coater instrument at a 100 mm/min withdrawal rate. To avoid confusion in the paper, the nomenclature given in Table 1 will be used throughout the paper.

**Table 1.** The nomenclature of the samples followed throughout the paper.

| Untreated AA2024 | PEO Treated | PEO + SG (TEOS + GPTMS) | PEO + SG-MT (TEOS + GPTMS + MTES) | PEO + SG-AP (TEOS + APTES) | PEO + SG-ZT (ZTP + MAPTMS + TEOS) |
|---|---|---|---|---|---|
| Substrate | PEO | PSG | PSG-MT | PSG-AP | PSG-ZT |

Dry sliding tests were performed on uncoated and coated specimens at room temperature using a Bruker reciprocating sliding tribometer and a 6 mm counter body of alumina. Keeping the stroke length of 5 mm and frequency of 5 cycles/s constant, 2 sets of conditions were imposed on all specimens under ambient conditions:

(a)  2.5 N load, 1200 s (20 min) sliding duration, 60 m distance;
(b)  3 N load, 2400 s sliding duration, 120 m distance.

Two tests were carried out in each set of conditions to ensure the reproducibility of results. Tangential frictional forces were recorded to calculate the friction coefficient (COF) continuously as a function of the sliding duration by means of a load cell. The average friction coefficient was calculated from the steady state area of the graph and later averaged again over two experiments.

Both a HIROX (Tokyo, Japan) KH-870 digital optical confocal microscope and a Hitachi (Tokyo, Japan) SU8020 Scanning Electron Microscope (SEM) were used for the surface analyses of wear traces after each test. Surface roughness was measured according to ISO 4287 standard procedure. Resistance to wear and debris under a tribo-contact was assessed using confocal microscopy for wear scar depths and width of wear traces. Wear volume loss was quantified through 2D area multiplied by stroke length by averaging three profiles on each wear trace. The specific wear rate, K (mm$^3$/N·m), is computed through the following equation (Equation (1)):

$$K = \frac{V}{F \times d} \tag{1}$$

where V is worn volume loss (mm$^3$), F is the normal load (N), and d is the reciprocating sliding distance (m). Consequently, the average specific wear rate was determined from two repetitions of tests. sol–gel traces were analyzed under SEM using Hitachi SU8020 equipment along with energy-dispersive X-ray spectroscopy (EDS). Color mapping of elemental composition was obtained through EDS equipped with Thermo Fisher Scientific (Waltham, MA, USA) Noran System 7 detector; 3D wear scars' profiles were drawn by P16+ profilometer from KLA Tencor (Milpitas, CA, USA).

## 3. Results and Discussion

### 3.1. SEM Analysis

The surface topography of the PEO-coated samples has been visualized via SEM images as depicted in Figure 1. The characteristic porous features of the PEO layer are caused by the repetitive melting and solidifying of the oxide layer during the process in the silicate-containing solution. Unnumbered pores with a diameter ranging from 1 to 10 μm are randomly placed throughout the surface as a result of the dielectric breakdown, plasma reactions, and the generation of such sparks. It can be noted that the trace of PEO pores and cracks has been diminished after the application of sol–gel coatings. Particularly, in the case of PSG-ZT, the formation of a layer resulted in the coverage of the porous layer in a way that flaws could barely be detectable anymore.

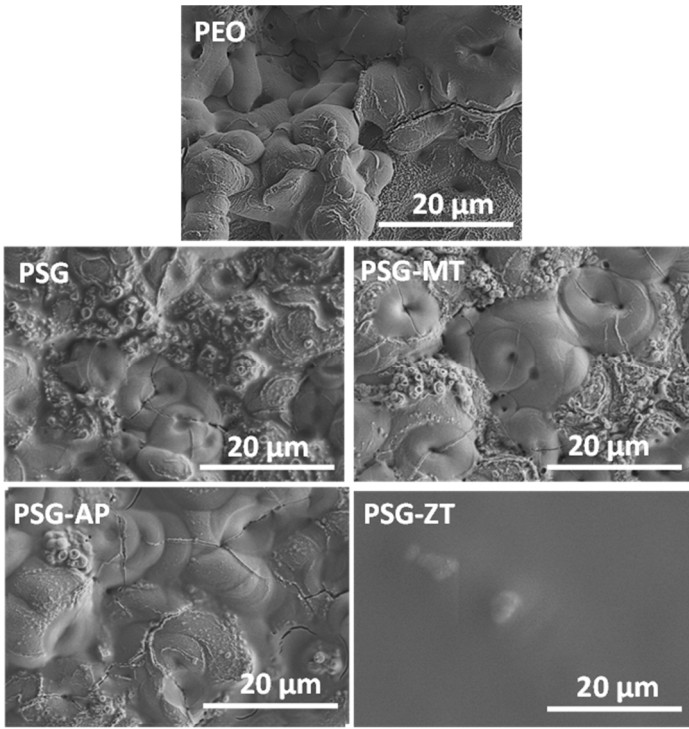

**Figure 1.** SEM analysis on surface morphology at 2.00 k magnification of coated samples.

The cross-section images along with the EDS mapping of the PEO/sol–gel coating systems are shown in Figure 2. The PEO oxide layer, having a thickness of $19.5 \pm 5$ μm, was produced by the combination of the substrate elements and the silicate solution, which is why EDS mapping images reported Al, O, and Si elements. Nevertheless, sol–gel sealing can be validated by the presence of the Si element inside the pores in the PEO/sol–gel samples compared to the distinct PEO sample. It was well-documented that the GPTMS precursor could provide the pore-filling ability to a sol–gel layer [32]. Accordingly, the almost similar sealing ability of the PSG and PSG-MT can be noticed by the EDS mapping. The PSG-AP sample contained some pores, demonstrating that the SG-AP was unable to effectively penetrate and seal the PEO pores. It appeared that the SG-AP was not likely to permeate through the pores and precipitate over the PEO surface. On the contrary, in the case of PSG-ZT, not only could a desirable pore filling be observed but also a homogenous layer over the PEO coating. In accordance with the top-view observation, the homogenous layer (with a thickness of $6.7 \pm 2$ μm) on top of PEO is recognized in the PSG-ZT.

The roughness measurement was performed and reported in Table 2. $R_a$ and $R_z$ corresponded to the arithmetical mean deviation and maximum height of the profile, respectively. The roughness of the PEO was reduced after the application of any sol–gel sealing, relating to the filling of the hollows with the sol–gel coating. Specifically, the roughness of the PSG-ZT was the minimum because of the ability of the SG-ZT sol–gel to not only seal the pores but also create a homogenous layer over the PEO layer. The roughness values of PSG and PSG-MT are somehow similar, illustrating identical sealing properties. For the PSG-AP, the $R_a$ is a little bit lower than PSG and PSG–MT, reflecting the dominant deposition of the SG-AP rather than the sealing ability as was also depicted in the SEM cross-section images.

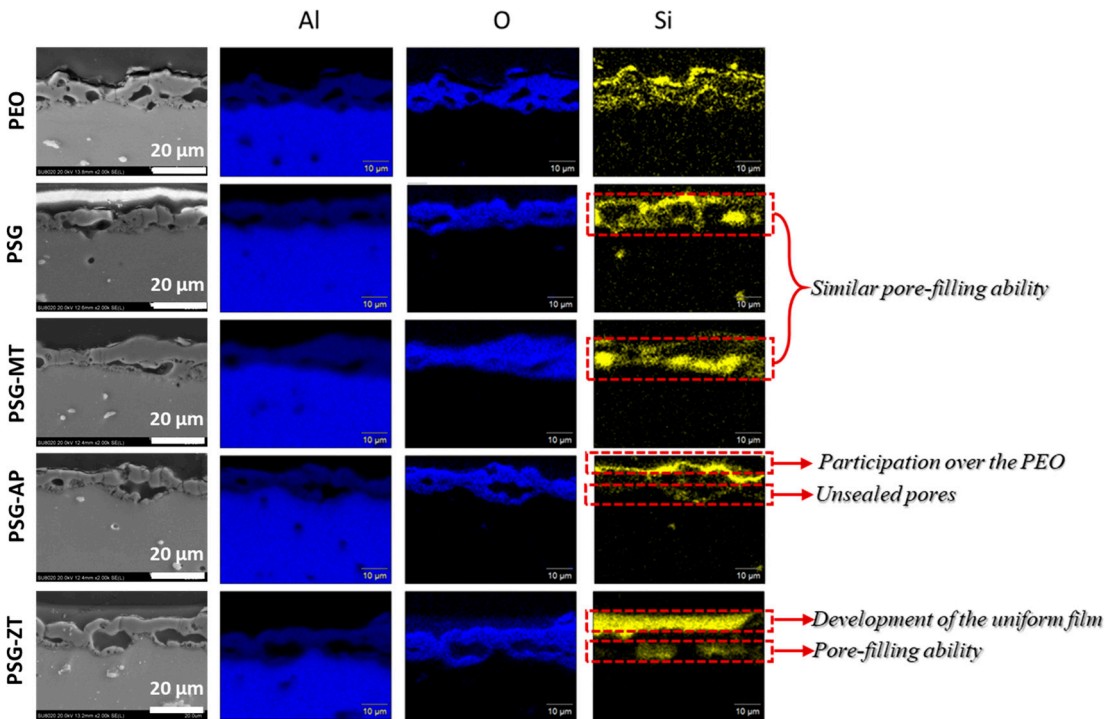

**Figure 2.** SEM cross-sectional analysis of coatings at 2.00 k magnification and color map.

**Table 2.** Roughness measurement results of different PEO/sol–gel coating systems.

| Average COF | PEO | PSG | PSG-MT | PSG-AP | PSG-ZT |
|---|---|---|---|---|---|
| $R_a$ | 1.7 | 1.3 | 1.3 | 1.2 | 0.2 |
| $R_z$ | 11.3 | 8.9 | 9.2 | 6.4 | 0.9 |

### 3.2. Tribological Evaluation

The variation of the friction coefficient (COF) with different surface treatments applied on the AA2024 alloy is illustrated in Figure 3. The COF for the substrate is selectively plotted in both conditions—(a) 2.5 N, 20 min, and (b) 3 N, 40 min—till the first highest peak and was snubbed further due to the high noise-to-signal ratio making the rest of the data unclear. The PEO-coated sample has shown a gradual increase in the coefficient of friction for Case (a), because when the loose porous structure of the PEO coating is being pressed when in contact during the wear test, it gradually exposed patches of the substrate from underneath with an average COF of 0.53. However, for Case (b), the PEO coating could not achieve a steady state as it failed to bear the load and produced more debris than the bare substrate against the alumina counter body, which led to a higher COF.

For the different sol–gels tested, PSG depicted a gradual increase in COF due to the running-in period and then stabilized for the rest of the tests with an average COF of 0.41 and 0.49 for Case (a) and Case (b), respectively. The initial gradual increase in COF is the result of a thin PSG coating, as it has a small running-in phase and helped to achieve a fast, steady state. A lubricating layer between two moving bodies known as the tribo-layer consisting of broken polymeric chains is formed. These polymeric chains being removed from the uppermost layer of the sol–gel are sheared and aligned in the direction of sliding in the contacting area between the substrate and the counter body. Hence, the third body tribo-layer formation proves that PSG has provided a good-enough sealing of the surficial defects with precursors (TEOS + GPTMS)—by making the network extendable through all hydrolyzable groups [29,33]—and when the network clusters of TEOS and GPTMS are aligned between contacting bodies after the running-in period, they facilitate the easy sliding by reducing shear stresses. Moreover, the PSG network holds well against the two

different loads (2.5 N and 3 N) by achieving a steady state rather quickly (after ~100 s) with an average COF of 0.41 and 0.49 for Cases (a) and (b), respectively. PSG-MT provided a trend similar to PSG in the COF curve at the milder conditions with an average of 0.40 for Case (a). However, a clear difference is noted in Case (b) where the sol–gel was not able to remain in the PEO pores at a higher load for a longer duration due to its less dense structure owing to the presence of the MTES precursor [34], which influences the compactness of the coating. The steady state only lasted for half of the test duration with an average COF of 0.50. Resultingly, when the sol–gel sealing was damaged, it exposed the PEO from underneath, which caused higher debris in the tribo-contact, hence the higher noise and poor lubrication. PSG-AP, owing to its -NH$_2$ precursor which was not able to fill the surface defects of the PEO coating, results in the inefficient sealing capability of the PEO layer [31,35]. Therefore, under load, it absorbs humidity from the surroundings and is pulled out from the superficial defects and exposed the PEO coating, showing a COF similar to the unsealed PEO coating with more debris and higher noise in the data. In the PSG-AP curves for both Cases (a) and (b), it can be noted that it is relatively difficult to obtain a steady-state range. However, the COF was averaged in the initial period having values of 0.55 in both cases with difficulty stating if the obtained COF value is due to the PSG-AP or PEO exposure. Out of all sol–gels, PSG-ZT had shown stable behavior with the longest steady-state range and the lowest average COF of 0.34 and 0.43 for both Cases (a and b), respectively. Instead of a gradual increase during the running period, a sudden increase was noticed but it still remained the lowest COF among all the sol–gels. The initial different behavior is attributed to the presence of a sol–gel layer with promising compactness properties as a top layer on the PEO layer [36]. Overall, sol–gel-sealed PEO coatings have shown promising results by effectively filling the surface defects as shown in Figure 2. The average COFs for all coatings in the steady-state range are presented in Table 3, and are in agreement with the research findings of Sopchenski-Santos et al. [28], where the bare AA2024, PEO coating, and sol–gel-sealed PEO coating were tested tribologically in a pin-on-disk tribometer at 1 N load against alumina ball.

A wear trace analysis of profile depths for the uncoated and coated samples are presented in Figures 4 and 5. The wear scar depth of PEO (Figure 4c) is much smaller than that of the bare substrate (Figure 4a) as the structure is able to withhold the load in contact for milder conditions in Case (a); nonetheless, it can be noted in Figure 4b,d that the PEO structure, owing to its relatively poor mechanical properties—since it has open, unfilled pores (Figure 2)—has, however, exposed the substrate.

The average specific wear rate is plotted, showing a significant decrease in wear rates for PEO-treated surfaces as compared to the bare substrate. This effect is more dominant in Case (a) with mild conditions, but, in Case (b), PEO, being a loose structure, is completely removed and the wear rate is close to that of the substrate. Similar outcomes were noticed by Sieber et al. [37] where PEO-coated commercial aluminum alloys showed a decrease in wear mass loss in comparison to the bare substrate.

For sol–gel-sealed specimens, a significant decrease is noted in the wear scars' depths as compared to the unsealed PEO coatings, particularly in PSG and PSG−ZT in Case (b) from 0.704 mm$^3$/Nm to 0.05 and 0.06 mm$^3$/Nm, respectively, as revealed in Figure 5 and Table 4. The sharp asperities of the PSG−coated structure have been flattened in the wear trace and look less like debris production but more like asperities pushed back into the structure as presented in Figure 5a for Case (a), and this phenomenon is more evident in Case (b) as displayed in Figure 5b. The good sealing characteristics of PSG due to four hydrolysable groups in TEOS and epoxy groups in GPTMS have helped to reduce the wear rate by a greater degree. As shown in Table 4, PSG has the lowest wear rates in both Cases (a) and (b). One thing to be noted here is that, in extreme conditions, the wear rate is further reduced by the densification of a hybrid inorganic/organic network where a higher load for a longer time has caused compactness in the surface without producing debris. This compression in the structure has caused mechanical strength in the coating, and hence it has maintained the coating integrity even better on higher loads. Contrarily, even with

a good sealing capability (with the MTES precursor), the structure of PSG−MT is not as compact as that of PSG. It is noted that the extent of damage after the wear test is higher in PSG−MT; this difference can be realized clearly in Figure 5d more than in Figure 5c for Case (b) and (a), respectively. At a low load (2.5 N) for a smaller period (1200 s), a minor pullout of sol–gel is observed from the PEO defects, hence the wear by the adhesive particles and sol–gel pullout. At the higher load (3 N for a longer period (2400 s)), the surface topography demonstrated (Figure 5d) a complete coating removal as the comparatively less compact structure does not withhold this load, and the coating is completely removed by the counter body penetration in the substrate. The adhesive wear mechanism can be explained by the high hardness of the alumina ball compared to the relatively soft coating which results in material transfer to the ball as a result, and, when exposed to air, the transferred material with free bonds oxidizes into hard particles and acts as the third body in the tribo-contact which causes more debris. In this case, wear resulted not only from third body rolling but also from PEO debris, and, ultimately, substrate exposure.

Figure 5e,f represent the surface profiles of PSG−AP for Case (a) and Case (b), respectively, and portray the worst wear resistance by producing the largest volume of material removal among all sol–gel−treated surfaces. For milder conditions, it illustrates the selective removal of material, leaving deep grooves inside the sol–gel network, with the possibility to have exposed PEO from underneath. However, for Case (b), the complete removal of the coating is rendering it inefficient in providing the necessary resistance to wear. Lastly, PSG−ZT has followed a behavior similar to that of PSG, presenting the lowest specific wear rate in extreme conditions (Case (b)). In Figure 5g, a small volume of material was removed during the running-in period as this is the thickest sol–gel layer evidenced by the cross-section analysis in Figure 2. For having a smooth run with lubricating contact, firstly, PSG−ZT flattens the asperities when under contact, and then the dense compact structure with the presence of ZTP holds against the load, leaving the coating resistant to high wear. Identical to PSG in Case (b), PSG−ZT has shown a decrease in wear rate by half as depicted in Figure 5h. This can be the result of the cement−like presence of cage−like and ladder-like siloxane structures inside the PEO pores as well as the evolution of a homogenous sol–gel layer over the PEO coating. Instead of shearing the weaker bonds among the layers of sol–gel, it compresses and maintains their structure against the load and generates lesser debris. The average specific wear rate of untreated and treated surfaces is condensed in Figure 6 and Table 4 with PSG and PSG−ZT ensuring good sealing and enhanced resistance on higher load, whereas PSG−MT and PSG−AP are associated with poor wear characteristics, respectively. In any case, the wear rate for sol–gel sealed coatings is much lower than that for the untreated substrate and PEO coatings.

**Table 3.** Average COF values in the steady-state range for Case (a) (2.5 N and 20 min) and Case (b) (3 N and 40 min).

| Average COF | Substrate | PEO | PSG | PSG-MT | PSG-AP | PSG-ZT |
|---|---|---|---|---|---|---|
| Case (a) 2.5 N, 20 min | NA | 0.53 | 0.41 | 0.40 | 0.55 | 0.34 |
| Case (b) 3 N, 40 min | NA | NA | 0.49 | 0.50 | 0.55 | 0.43 |

**Table 4.** Average specific wear rate for different surface treatments at (a) 2.5 N load for 20 min, and (b) 3 N load for 40 min.

| Average Specific Wear Rate $mm^3/(Nm)$ | Substrate | PEO | PSG | PSG-MT | PSG-AP | PSG-ZT |
|---|---|---|---|---|---|---|
| Case (a) | 1.08728 | 0.16073 | 0.07049 | 0.09131 | 0.16482 | 0.11602 |
| Case (b) | 0.85314 | 0.74016 | 0.05883 | 0.40587 | 0.45725 | 0.06379 |

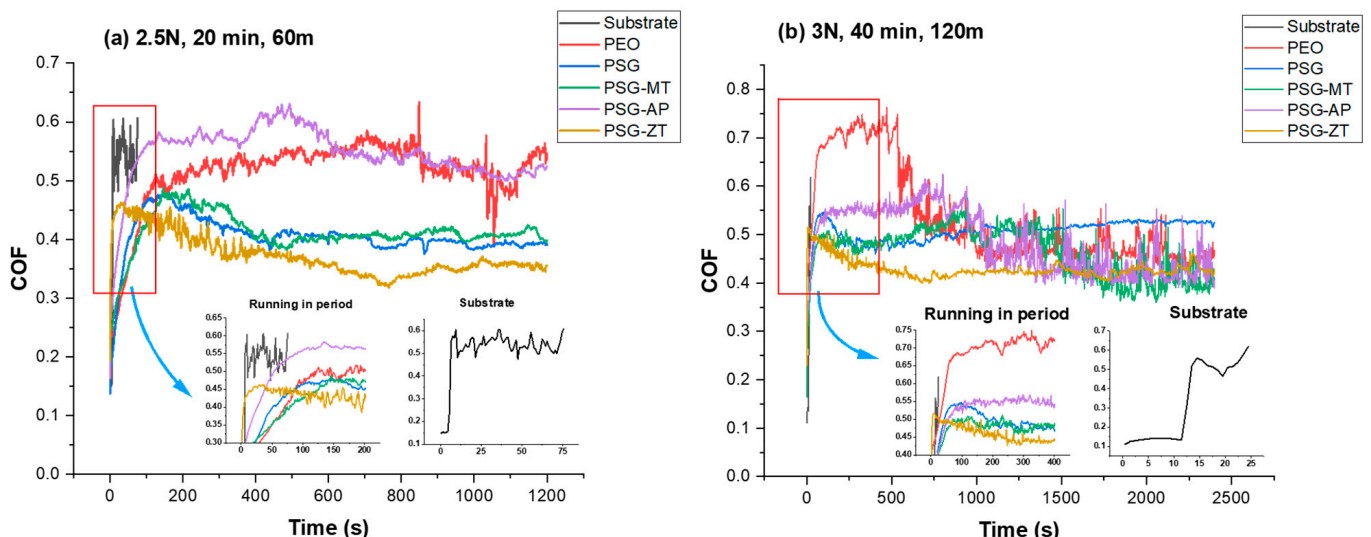

**Figure 3.** COF of tested sol–gels at (**a**) 2.5 N load and 20 min sliding, and (**b**) 3 N load for 40 min sliding time.

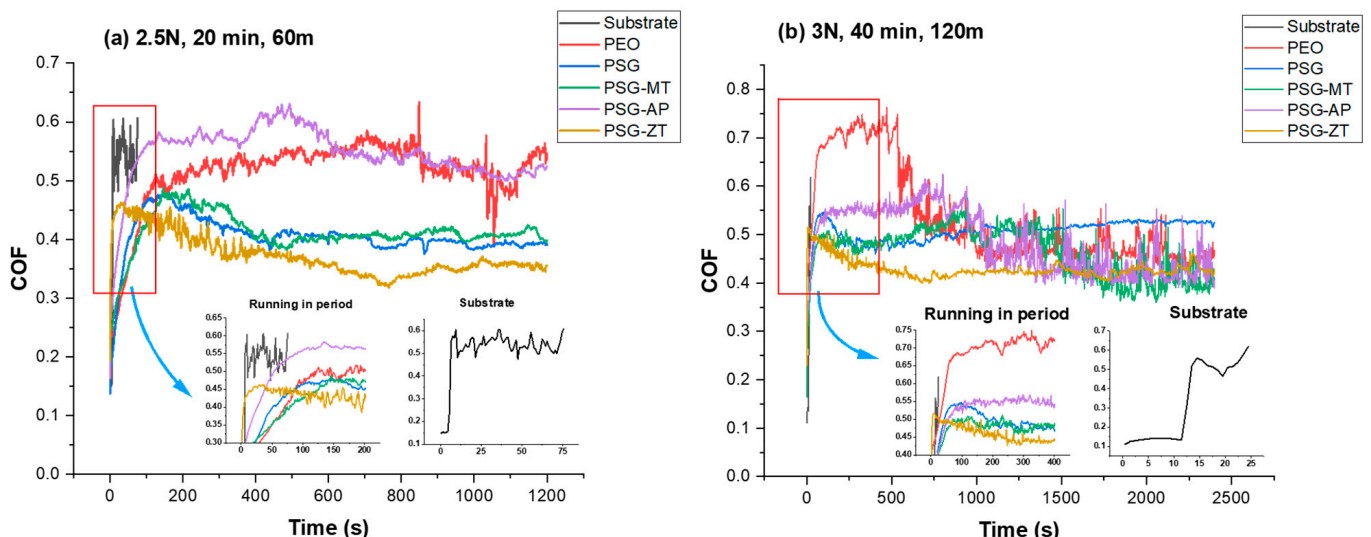

**Figure 4.** Mechanical profilometer 3D surface scan: (**a**,**b**) for bare AA2024 substrate, and (**c**,**d**) for PEO–coated specimens at 2.5 N load for 20 min and 3 N load for 40 min, respectively.

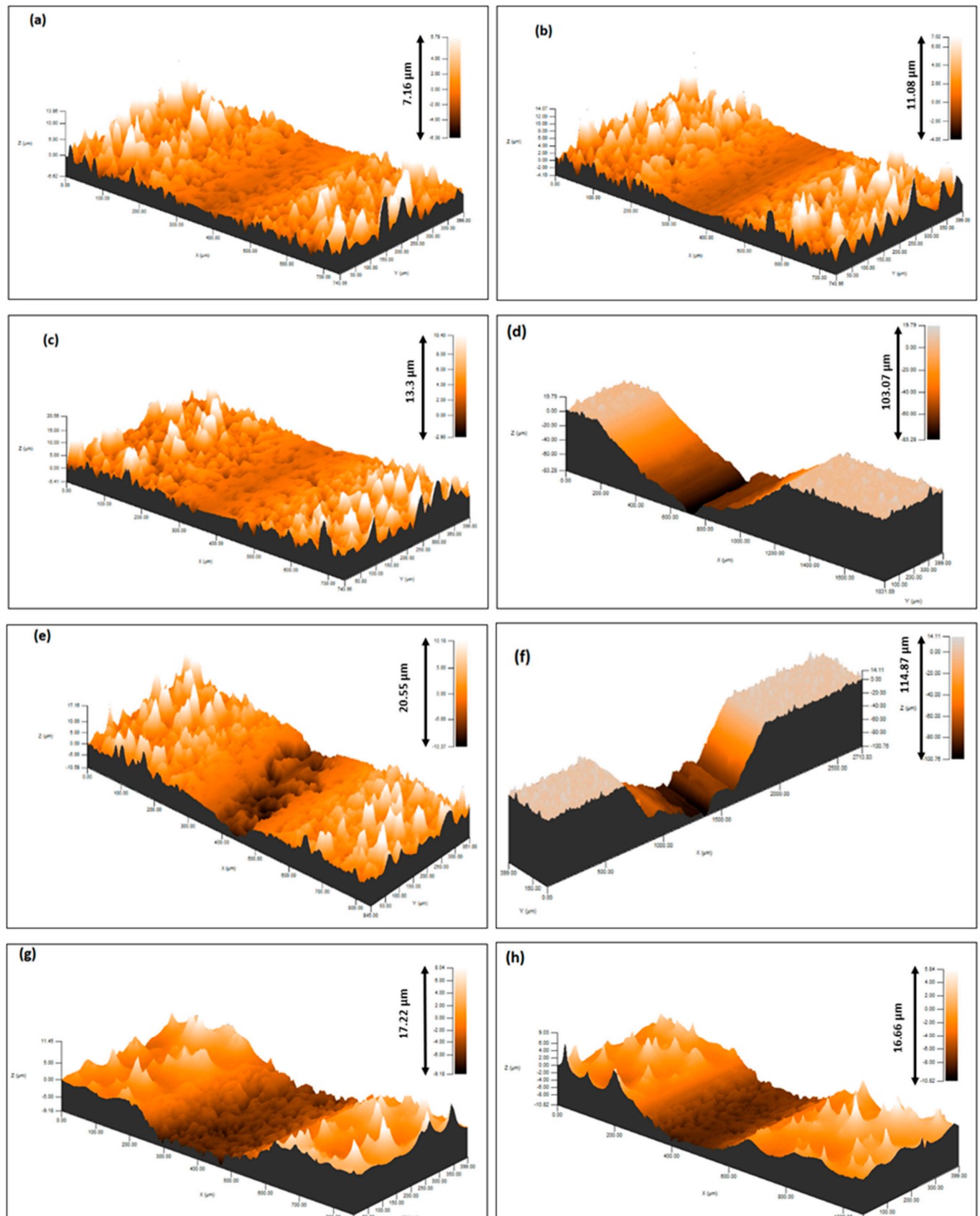

**Figure 5.** Mechanical profilometer 3D scan: (**a**,**b**) for PSG, (**c**,**d**) for PSG−MT, (**e**,**f**) for PSG−AP, and (**g**,**h**) for PSG−ZT, tested at 2.5 N load for 20 min and 3 N load for 40 min, respectively.

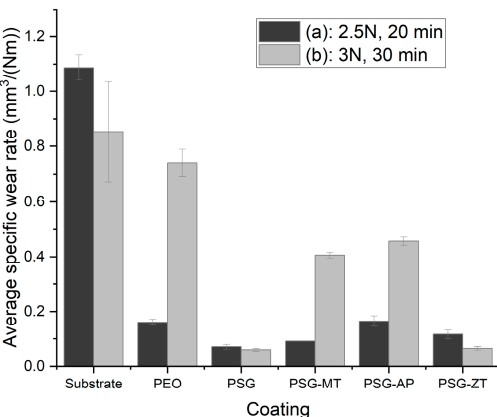

**Figure 6.** Graphical representation of average specific wear rate at (**a**) 2.5 N load for 20 min, and (**b**) 3 N load for 40 min for different surface treatments.

The typical distribution of elements on the wear traces of coated specimens is presented in Figure 7. It can be noted that all considered elements are distributed uniformly outside the wear trace for all sol–gel-sealed PEO coatings, ensuring uniform coating application [38]. Al is present in the wear tracks of all coatings. O exposure from the PEO coating underneath the sol–gel layer is proportional to the wear rate obtained in Figure 6. PSG and PSG-ZT exhibited a narrower removal of the sol–gel layer supporting their wear rates. With more presence of Si in the PSG wear track, it looks like PSG has a wider but shallower wear track than PSG-ZT. However, PSG-ZT has a narrower and relatively deeper wear trace. All in all, for wear scars obtained from the test at milder conditions (Case (a)), Si is still present in the wear traces of sol–gel layers of all coated samples. This means the wear test did not remove the sol–gel layers, proving the good adhesion of the coating [28].

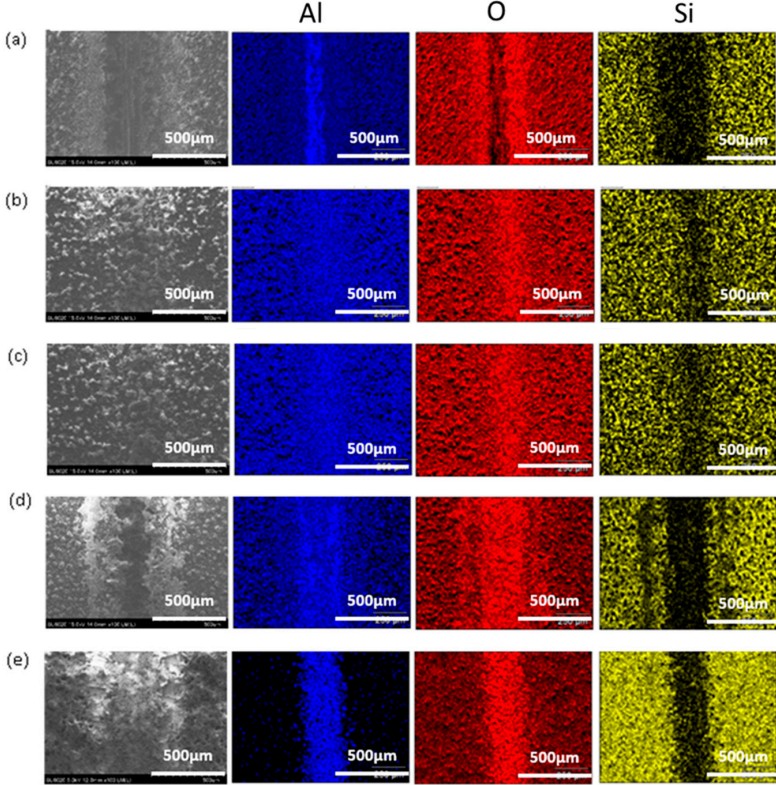

**Figure 7.** EDS map obtained at 100× through SEM image of coatings after wear test at 2.5 N load for 20 min: (**a**) PEO coating, (**b**) PSG, (**c**) PSG-MT, (**d**) PSG-AP, and (**e**) PSG-ZT.

In extreme conditions (3 N load for 40 min), it is clear that the wear tracks are wider in comparison to Figure 7. Specimens with unsealed PEO (Figure 8a), PSG-MT (Figure 8c), and PSG-AP (Figure 8d) show a complete absence of Si in the wear track in the color maps, indicating the pullout of sol–gel constituents from PEO pores as well as exposed substrate by the removal of the PEO layer. In addition to Si from the electrolyte, PEO coatings synthesized in an alkaline environment are primarily composed of $\alpha$-Al$_2$O$_3$ and $\gamma$-Al$_2$O$_3$ [39,40]. Therefore, the depletion of O and Si elements and the high content of Al in the EDS color maps from the wear trace are proof of the Al substrate exposure and the removal of PEO layers as well. On the contrary, in PSG (Figure 8b) and PSG-ZT (Figure 8e), no parallel lines are observed, meaning that wear did not occur by abrasion. Similar results were obtained by Javadi et al. on the AA2024 alloy [25]. These two sol–gels have proven to be a good cementing agent in the sealing of PEO pores, as only a minute pullout of Si-containing species is noticed, and thus has good adhesive properties. By creating a dense structure with the PEO structure, it has improved the mechanical properties as confirmed by Pezzato et al. [41].

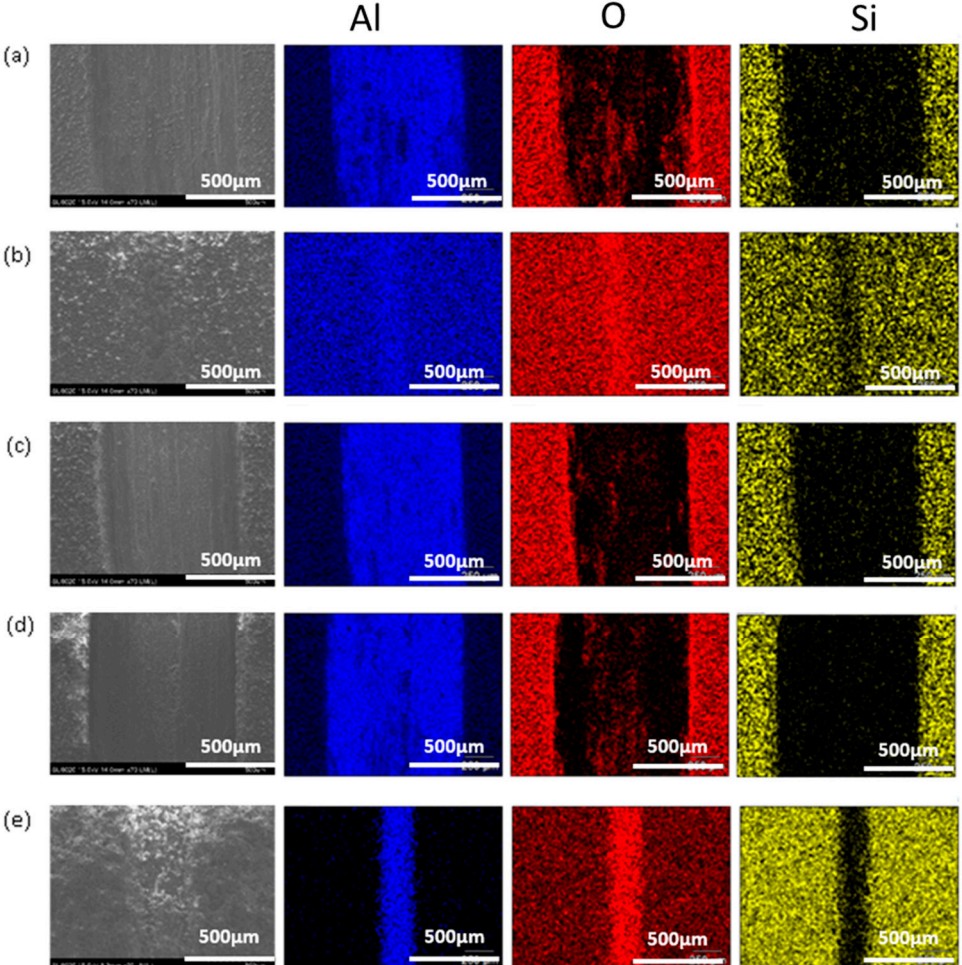

**Figure 8.** EDS map obtained at 70× through SEM image of coatings after wear test at 3 N load for 40 min: (**a**) unsealed PEO, (**b**) PSG, (**c**) PSG−MT, (**d**) PSG−AP, and (**e**) PSG−ZT.

Figure 9a indicates the transfer of the PEO coating from the wear track to the alumina counter body for unsealed PEO coatings, and this effect is more dominant with a higher coating transfer in Case (b). The transferred coating onto the ball could form hard abrasive particles by agglomeration, hence accelerating the wear phenomena. A similar trend is observed in PSG−MT and PSG−AP as the precursors MTES and APTES have a comparatively poor sealing performance due to a less dense network facilitating pullout during

the wear test, once the coated material transfer to the ball began after the sol–gel being pulled out of the PEO. Along with a high hardness, alumina is known for its brittle nature which limits its mechanical properties [42], resulting in the production of third body rolling brittle particles in the tribo-contact and, hence, more debris production. The PSG−MT and PSG−AP wear tracks are wider, with the complete removal of PEO coating at a higher load as well. However, PSG and PSG−ZT show no material transfer on the ball even at higher loads because of decreased shear stresses and the efficient sealing of PEO pores owing to their high branching [33] and organic groups present in the sol–gel structure [36].

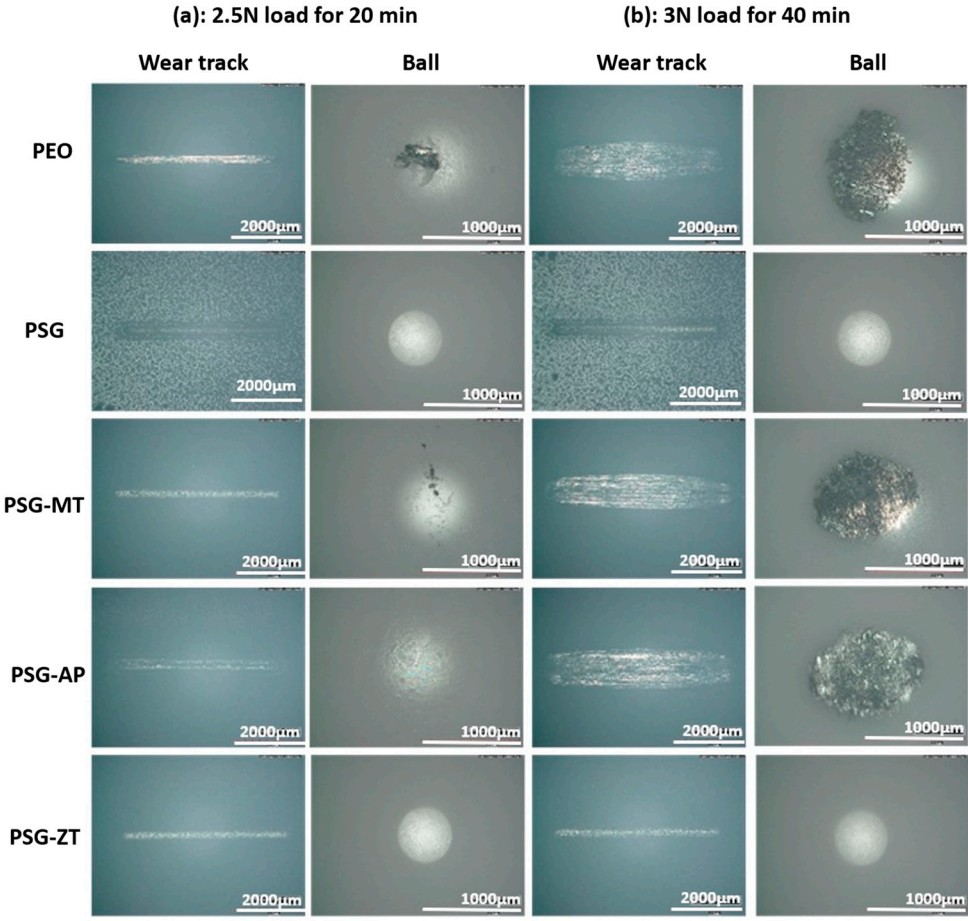

**Figure 9.** Optical microscopy images of wear tracks and counter body at (**a**) 2.5 N load for 20 min and (**b**) 3 N load for 40 min.

## 4. Conclusions

In the present work, a tribological evaluation of the previously developed sol–gel formulations have been carried out. With the aim of investigating the tribo−layer and self-healing capability of the sol–gel sealing of the PEO layer on the AA2024 alloy, i.e., coefficient of friction, the wear loss volume, elemental traces of sol–gel in the wear tracks, transfer of wear debris onto the counter body, wear track depths, and asperity profiles have been studied. The following conclusions can be drawn from the experimental results:

- The average COF values in the steady-state range for Case (a) (2.5 N and 20 min) and Case (b) (3 N and 40 min) are, respectively, followed as 0.53 and NA for PEO, 0.41 and 0.49 for PSG, 0.40 and 0.50 for PSG−MT, 0.55 and 0.55 for PSG−AP, and 0.34 and 0.43 for PSG−ZT.
- The average specific wear rate in 3 N load for the 40 min condition was 0.74016, 0.05883, 0.40587, 0.45725, and 0.06379 mm$^3$/(Nm), for PEO, PSG, PSG−MT, PSG−AP, and PSG−ZT, respectively.

- The presence of GPTMS along with TEOS in PSG has created a dense sol–gel network in the PEO structure that has a good penetrative ability to seal the PEO layer resulting in a low COF and wear loss volume. Similarly, PSG−ZT, due to a higher sol–gel content, exhibited the same behavior. In fact, the more the hydrolyzed groups inside of a sol–gel network, the more opportunity for either the network formation or chemical adsorption of a sol–gel cluster to the PEO layer. The addition of MTES to the sol–gel formulation brought about a lower number of hydrolyzed groups possessing an un-hydrolyzed methyl functional group. Not only was the compactness of the PSG coating higher than that of PSG−MT but also its wettability, leading to the creation of more chemical bonds to the oxide groups over the PEO sample. For the PSG−ZT formulation, a high content of sol–gel precursors were utilized, which, interestingly, resulted in the formation of a dense layer over the PEO along with pore-filling ability. Hence, one could expect that, even though a high content of organic compounds was employed to obtain PSG−ZT, the wettability could be comparable with PSG.
- On the higher loads for PSG and PSG−ZT, they showed lesser wear due to the structure of the sol–gel layer being pushed together and aligning in the direction of sliding, giving lubricating characteristics to the surface.
- Moreover, the relatively low amount of wear debris in PSG and PSG−ZT with no material transfer to the counter body is indicative of the improved compactness of the coating and adhesion of the coating. Supporting EDS color maps through the SEM analysis shows the presence of sol–gel constituents in the wear tracks after the test.
- PSG−MT and PSG−AP are rendered inefficient to enhance the mechanical properties of the coatings, especially in severe conditions (Case (b)). In the case of PSG−AP, the insufficient sealing ability and the configuration of the hydrophilic amine group over the PEO sample could be the reason for such behavior. In any case, sol–gel-treated PEO layers have shown tremendous improvement in tribological properties compared to untreated samples.

**Supplementary Materials:** The following supporting information can be downloaded at: https://www.mdpi.com/article/10.3390/coatings13050871/s1, Table S1: Composition of the sol–gel used to seal PEO pores through dip coating.

**Author Contributions:** H.A.K.: investigation, methodology, validation, and writing—original draft. S.A.: investigation, methodology, validation. Y.P.: investigation, validation. V.V.: writing—review & editing, funding acquisition, and supervision. M.-G.O.: conceptualization, methodology, validation, writing—review & editing, supervision, and funding acquisition. All authors have read and agreed to the published version of the manuscript.

**Funding:** This work is funded by the 2018 SEALCERA project (Fédération Wallonie Bruxelles) as part of the ARC (Action de Recherche Collective). Ayesha Khalid wishes to thank the FRIA (Fonds pour la Formation à la Recherche dans l'Industrie et l'Agriculture) for funding.

**Institutional Review Board Statement:** Not applicable.

**Informed Consent Statement:** Not applicable.

**Data Availability Statement:** The data presented in this study are available upon request from the corresponding author.

**Acknowledgments:** The University of Mons provided financial support for the 2018 SEALCERA project (Fédération Wallonie Bruxelles) as part of the ARC (Action de Recherche collective), which the authors sincerely acknowledge. Ayesha Khalid wishes to thank the FRIA (Fonds pour la Formation à la Recherche dans l'Industrie et l'Agriculture) for funding. Last but not least, special thanks to Xavier Buttol and Dominique Hautcoeur for performing the roughness measurement at the Belgian Ceramic Research Center (CBRC).

**Conflicts of Interest:** The authors declare no conflict of interest.

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
