# Peer review of "Comparison of Tribological Characteristics of AA2024 Coated by Plasma Electrolytic Oxidation (PEO) Sealed by Different sol–gel Layers"

_coatings, doi:10.3390/coatings13050871_

Round 1
Reviewer 1 Report
The article describes the results that are valuable for science and practice, but there are several questions and recommendations:
1. I recommend to describe in more detail the regularities of the influence of electrical modes (frequency, shape, duty cycle and strength of electric current) on the structure and mechanical properties (corrosion resistance, thickness, porosity, wear resistance, microhardness) obtained PEO coatings on the surface of aluminum alloy and obtain a physical model of coating formation based on experimental data.
2. Perhaps it was worth writing down to create a mathematical model describing the dependence of mechanical properties (thickness and microhardness) plasma electrolytic oxidation (PEO) coatings on the surface aluminum alloy from the modes of the investigated process.
3. The fonts in Figs. 4,5 are very small for its full analysis.
4. On fig. 1 shows the surface morphology of an aluminum alloy AA2024 coated by plasma electrolytic oxidation (PEO). I suggest using images at some multiple magnifications. This will allow a more detailed understanding of its structure.
5. I propose to remove reference [21] from the conclusions. References in conclusions are unconventional. The conclusions should show the value of the NEW results obtained by the authors for science and practice, be clear, understandable, structured.
6. I propose to add to the introduction a review of a recent article in this scientific direction:
https://www.sciencedirect.com/science/article/pii/S2238785421007456
Author Response
The article describes the results that are valuable for science and practice, but there are several questions and recommendations:
- I recommend to describe in more detail the regularities of the influence of electrical modes (frequency, shape, duty cycle and strength of electric current) on the structure and mechanical properties (corrosion resistance, thickness, porosity, wear resistance, microhardness) obtained PEO coatings on the surface of aluminum alloy and obtain a physical model of coating formation based on experimental data.
Response: The source of power mode employed in the PEO process has an impact on the hardness, growth rate, phase composition, structure, morphology, and degree of porosity of the coatings. PEO treatment operating in DC mode results in coatings with a lower oxide development rate and higher porosity due to its restricted control and difficulty in changing discharge characteristics. Nevertheless, pulsed DC mode allows the opportunity to regulate the discharge duration and can potentially use less energy. By using AC mode, electrode polarization is avoided, and arc interruption may be used to conveniently manage the process. In comparison to coatings created using DC, AC, and unipolar pulsed modes, the bipolar pulsed mode greatly enhances coating characteristics, drawing the attention of numerous researchers. Since it lessens the frequency of strong plasma discharges and high temperature spikes during the PEO process, dense coatings with superior corrosion resistance and larger coating thickness may be created in bipolar pulsed current mode. Under AC or bipolar-pulsed regimes, frequently using frequencies up to several thousand Hz and variable duty cycles, a three-layered morphology (a submicron thick, homogeneous, dense barrier layer, an intermediate relatively compact micro-cracked layer rich in α-alumina, and an outer porous relatively loose layer rich in electrolyte derived compounds, e.g., SiO2) is obtained. According to theory, increasing the quantity of α-Al2O3 improves the wear performance of PEO coatings. Khan et al. discovered that a reduced duty cycle resulted in a comparable decline in the ratio of α-Al2O3 to γ-Al2O3 in PEO coatings on 6082 aluminum alloy generated by the pulsed unipolar current. The concentration of α -Al2O3 in the coatings was observed to rise with the application of larger current densities and the lengthening of the deposition period, which produced thicker coatings. V. Dehnavi et al. found that increasing the pulse on time by employing a lower frequency and higher duty cycle generates micro discharges with lower spatial density but higher intensity which results in higher concentrations of Si-rich species on the surface of the PEO coatings. In general, high frequencies and negative pulses help to limit the duration of individual micro discharges, prevent their reappearance at the same location and transition into destructive arc discharges.
This explanation has been added to the manuscript.
- Perhaps it was worth writing down to create a mathematical model describing the dependence of mechanical properties (thickness and microhardness) plasma electrolytic oxidation (PEO) coatings on the surface aluminum alloy from the modes of the investigated process.
Response: This is indeed an excellent idea, but out of the scope of this study. Authors warmly acknowledge your suggestion which will be used in future works.
- The fonts in Figs. 4,5 are very small for its full analysis.
Response: The font in Fig.4 and Fig. 5 are not modifiable as the instrument generate the profiles. The authors tried to compensate the small size of fonts by putting a scale next to each one.
- On fig. 1 shows the surface morphology of an aluminum alloy AA2024 coated by plasma electrolytic oxidation (PEO). I suggest using images at some multiple magnifications. This will allow a more detailed understanding of its structure.
Response: Unfortunately, due to the lack of revision time, it is not possible to take new SEM images in different magnifications. In our previous investigation in which the same samples were considered for corrosion resistance properties, different magnifications have been put in the paper.
- I propose to remove reference [21] from the conclusions. References in conclusions are unconventional. The conclusions should show the value of the NEW results obtained by the authors for science and practice, be clear, understandable, structured.
Response: The reference [21] has been removed from the conclusion part.
- I propose to add to the introduction a review of a recent article in this scientific direction:
https://www.sciencedirect.com/science/article/pii/S2238785421007456
Response: V. Hutsaylyuk et al. reduced the unfavorable impact of hydrogen on the plasma electrolytic oxidation of aluminum alloys and accordingly increased the efficiency of the synthesis of PEO layers with high abrasive wear resistance.
This explanation has been added to the manuscript.

Reviewer 2 Report
Dear Authors,
I have read your paper "Comparison of tribological characteristics of AA2024 coated by plasma electrolytic oxidation (PEO) sealed by different sol-gel layers" carefully.
Explanations are clear and the review is easy to read.
However, it requires few corrections.
- Please add, abbreviation of the COF in the abstract..
- Please add, the informations about roughness of the coatings.
- Please add, general quantitative results in the conclusions.
The paper can be accepted for publication only after minor improvements.
Author Response
Dear Authors,
I have read your paper "Comparison of tribological characteristics of AA2024 coated by plasma electrolytic oxidation (PEO) sealed by different sol-gel layers" carefully.
Explanations are clear and the review is easy to read.
However, it requires few corrections.
- Please add, abbreviation of the COF in the abstract.
Response: The modification has been applied to the manuscript.
- Please add, the informations about roughness of the coatings.
Response: The roughness measurement was done and reported in Table 2. Ra and Rz are corresponded to the arithmetical mean deviation and maximum height of the profile, respectively. The roughness of the PEO was reduced after the application of any sol-gel sealing, relating to the filling of the hollows with the sol-gel coating. Specifically, the roughness of the PSG-ZT was the minimum because of the ability of the SG-ZT sol-gel to not only seal the pores but also create a homogenous layer over the PEO layer. The roughness values of PSG and PSG-MT are somehow similar, illustrating identical sealing properties. For the PSG-AP, the Ra is a little bit lower than PSG and PSG-MT, reflecting the dominant deposition of the SG-AP rather than the sealing ability as it was also depicted in the SEM cross-section images.
Table 2. Roughness measurement results of different PEO/sol-gel coating systems.
|
Average COF |
PEO |
PSG |
PSG-MT |
PSG-AP |
PSG-ZT |
|
Ra |
1.7 |
1.3 |
1.3 |
1.2 |
0.2 |
|
Rz |
11.3 |
8.9 |
9.2 |
6.4 |
0.9 |
This explanation has been added to the manuscript.
- Please add, general quantitative results in the conclusions.
Response: The modification has been applied to the manuscript
The paper can be accepted for publication only after minor improvements.

Reviewer 3 Report
The manuscript presents very interesting results of tribological research. The method of fixing the coating proposed by the authors is original. Despite this, when reading the text, the reviewer had questions regarding the presentation and interpretation of the results.
1. Page 2: At the first mention in the text, it is necessary to give a decoding of all abbreviations of the sol-gel precursors. References to the previously presented results of the same authors are not sufficient. And if these are precursors, what is the final result of the sol-gel synthesis?
2. Page 3, line 129: Check the spelling of the wear rate dimension and the corresponding formula. The multiplication sign must be present.
3. Page 4: Was the thickness of the homogeneous layer evaluated for all samples?
4. Page 5: What was the rationale for choosing a shorter tribological test time for a lower load?
Page 6: Explanations for the differences in the friction coefficient for different sol-gel coatings are not entirely clear without data on the chemical composition and thickness of the layers.
Similar remarks apply to the wear rate.
Author Response
The manuscript presents very interesting results of tribological research. The method of fixing the coating proposed by the authors is original. Despite this, when reading the text, the reviewer had questions regarding the presentation and interpretation of the results.
- Page 2: At the first mention in the text, it is necessary to give a decoding of all abbreviations of the sol-gel precursors. References to the previously presented results of the same authors are not sufficient. And if these are precursors, what is the final result of the sol-gel synthesis?
Response: The various sol-gel formulations including the decoding of all abbreviations were added to the experimental section.
- Page 3, line 129: Check the spelling of the wear rate dimension and the corresponding formula. The multiplication sign must be present.
Response: The modification has been applied to the manuscript.
- Page 4: Was the thickness of the homogeneous layer evaluated for all samples?
Response: The inherent porous structure of the PEO layer brought some pathways for sol-gels to diffuse and seal pores and cracks. As it was depicted by SEM images, the Si element clarified that PSG-ZT was the only coating system with a homogenous sol-gel top layer. The rest of the PEO/sol-gel coated samples either diffuse or partially precipitate over the PEO layer. All in all, the thickness of the PEO layer was determined as 19.5 ± 5 µm and the homogenous sol-gel layer in the PSG-ZT sample was 6.7 ± 2 µm.
Some notations were added to Figure 2 to clarify the sealing properties of various PEO/sol-gel coating systems.
- Page 5: What was the rationale for choosing a shorter tribological test time for a lower load?
Response: The thickness of the substrate is 1.5 mm. If a higher load is applied to the coated sample, the sample would be completely broken down. The thickness and load values lie somewhat in the same range as the following scientific works where the substrate was aluminum as well:
- Fu, J., Li, M., Liu, G., Ma, S., Zhu, X., Ma, C., ... & Yan, Z. (2020). Robust ceramic based self-lubricating coating on Al–Si alloys prepared via PEO and spin-coating methods. Wear, 458, 203405.
- Feng Su, J., Nie, X., Hu, H., & Tjong, J. (2012). Friction and counterface wear influenced by surface profiles of plasma electrolytic oxidation coatings on an aluminum A356 alloy. Journal of Vacuum Science & Technology A: Vacuum, Surfaces, and Films, 30(6), 061402.
As we’re comparing uncoated aluminum with PEO-coated aluminum, we chose the milder conditions because the wear of aluminum is already very important in those so it is an easy way to screen candidates' coatings systems. More in-depth tribological testing, with harsher conditions, is envisaged for a full characterization of the best coatings.
- Page 6: Explanations for the differences in the friction coefficient for different sol-gel coatings are not entirely clear without data on the chemical composition and thickness of the layers. Similar remarks apply to the wear rate.
Response: By comparing PSG and PSG-MT, even though a similar sealing ability was reported, the presence of one unreacted methyl group could reduce the compactness of the sol-gel network. In the PSG-AP, the coating not only was not compact enough but also the poor sealing ability made it the worst PEO/sol-gel coating system in both terms of corrosion resistance and mechanical properties. The creation of a uniform layer over the PEO, as well as penetration through the intrinsic pores, hydrophobic properties, and the evolution of the cage-like along with the ladder-like siloxane structure, made the PSG-ZT a compact and dense coating system. The lack of material transfer to the counter body and the comparatively low quantity of wear debris in PSG and PSG-ZT referred to increased coating compactness and adherence. SEM analysis supporting EDS color maps revealed the presence of sol-gel components in the wear tracks following the test. Particularly in more severe circumstances, PSG-MT and PSG-AP are proven ineffective in improving the mechanical characteristics of the coatings (case b). The arrangement of the hydrophilic amine group over the PEO sample and poor sealing ability in the case of PSG-AP, and the presence of unreacted methyl group resulting in the obtainment of lower compactness in PSG-MT could be responsible for this behavior All in all, compared to untreated samples, PEO layers that have been sol-gel treated exhibit significantly improved tribological characteristics.

Round 2
Reviewer 1 Report
Accept.
Reviewer 3 Report
The authors responded to all comments of the reviewer. Thank you.